# Evaluation of Potential Targets for Fluorescence-Guided Surgery in Pediatric Ewing Sarcoma: A Preclinical Proof-of-Concept Study

**DOI:** 10.3390/cancers15153896

**Published:** 2023-07-31

**Authors:** Bernadette Jeremiasse, Zeger Rijs, Karieshma R. Angoelal, Laura S. Hiemcke-Jiwa, Ella A. de Boed, Peter J. K. Kuppen, Cornelis F. M. Sier, Pieter B. A. A. van Driel, Michiel A. J. van de Sande, Marc H. W. A. Wijnen, Anne C. Rios, Alida F. W. van der Steeg

**Affiliations:** 1Department of Surgery, Princess Maxima Center for Pediatric Oncology, 3584 CS Utrecht, The Netherlands; b.jeremiasse-4@prinsesmaximacentrum.nl (B.J.); k.r.angoelal-2@prinsesmaximacentrum.nl (K.R.A.); m.a.j.vandesande-2@prinsesmaximacentrum.nl (M.A.J.v.d.S.); m.h.w.wijnen-5@prinsesmaximacentrum.nl (M.H.W.A.W.); a.f.w.vandersteeg@prinsesmaximacentrum.nl (A.F.W.v.d.S.); 2Department of Orthopedic Surgery, Leiden University Medical Center, 2333 ZA Leiden, The Netherlands; 3Department of Pathology, Princess Maxima Center for Pediatric Oncology, 3584 CS Utrecht, The Netherlands; l.s.jiwa-3@prinsesmaximacentrum.nl (L.S.H.-J.); a.m.deboed@prinsesmaximacentrum.nl (E.A.d.B.); 4Department of Pathology, University Medical Center Utrecht, 3584 CG Utrecht, The Netherlands; 5Department of Surgery, Leiden University Medical Center, 2333 ZA Leiden, The Netherlands; p.j.k.kuppen@lumc.nl (P.J.K.K.); c.f.m.sier@lumc.nl (C.F.M.S.); 6Department of Orthopedic Surgery, Isala Hospital, 8025 AB Zwolle, The Netherlands; p.b.a.a.van.driel@isala.nl; 7Research Department, Princess Maxima Center for Pediatric Oncology, 3584 CS Utrecht, The Netherlands; a.c.rios@prinsesmaximacentrum.nl; 8Oncode Institute, Jaarbeursplein 6, 3521 AL Utrecht, The Netherlands

**Keywords:** CD99, CD117, GD2, fluorescence-guided surgery, pediatric, Ewing sarcoma

## Abstract

**Simple Summary:**

The only cure for children with Ewing sarcoma (ES) is surgery. Unfortunately, surgeons are often not able to differentiate healthy from malignant tissue. Fluorescent imaging during the operation will facilitate recognition of malignant cells, but unfortunately there are no ES specific tracers available yet. We searched for proteins on ES cells that could be used as a target against which specific tracers could be developed. The most promising proteins, CD99, CD117, and GD2, were found in paraffin-embedded tissue samples collected from ES patients. Tracers against CD99 and CD117, consisting of monoclonal antibodies attached with a fluorescent dye, showed positive signals on cultured ES cells. In a proof-of-concept study, these tracers were topically applied on fresh ES tissue, showing a signal in the tumor. Our results indicate the applicability for fluorescence-guided surgery of ES-based tracers, but these data have to be confirmed in a larger cohort of pediatric ES patients.

**Abstract:**

Fluorescence-guided surgery (FGS), based on fluorescent tracers binding to tumor-specific biomarkers, could assist surgeons to achieve complete tumor resections. This study evaluated potential biomarkers for FGS in pediatric Ewing sarcoma (ES). Immunohistochemistry (IHC) was performed to assess CD99, CXCR4, CD117, NPY-R-Y1, and IGF-1R expression in ES biopsies and resection specimens. LINGO-1 and GD2 evaluation did not work on the acquired tissue. Based on the immunoreactive scores, anti-CD99 and anti-CD117 were evaluated for binding specificity using flow cytometry and immunofluorescence microscopy. Anti-GD2, a tracer in the developmental phase, was also tested. These three tracers were topically applied to a freshly resected ES tumor and adjacent healthy tissue. IHC demonstrated moderate/strong CD99 and CD117 expression in ES tumor samples, while adjacent healthy tissue had limited expression. Flow cytometry and immunofluorescence microscopy confirmed high CD99 expression, along with low/moderate CD117 and low GD2 expression, in ES cell lines. Topical anti-CD99 and anti-GD2 application on ES tumor showed fluorescence, while anti-CD117 did not show fluorescence for this patient. In conclusion, CD99-targeting tracers hold promise for FGS of ES. CD117 and GD2 tracers could be potential alternatives. The next step towards development of ES-specific FGS tracers could be ex vivo topical application experiments on a large cohort of ES patients.

## 1. Introduction

Ewing sarcoma (ES) is a rare, aggressive, small blue round cell tumor that can occur in bone or, more rarely, in soft tissue [1,2]. It is the second most common primary osseous malignancy in children and young adults, after osteosarcoma [3]. Diagnostic workup involves plain radiographs, magnetic resonance imaging (MRI), sometimes with additional computed tomography (CT), followed by a biopsy, and (if possible) molecular confirmation [1,4,5]. Once confirmed, subsequent ES treatment is highly individualized, but generally consists of multimodality therapies comprising surgery and (neo)adjuvant chemotherapy with or without radiotherapy [1,6,7]. Clinical outcomes are highly dependent on surgical margins, as incomplete margins negatively affect both local recurrence (LR) and overall survival (OS) [8,9,10]. Moreover, incomplete resections often necessitate postoperative radiotherapy to reduce the risk for LR, while radiotherapy increases the risk for local complications [11]. This is especially relevant in the context of pediatric patients, as survivors are susceptible to encountering various (long-term) treatment-related side effects, including compromised growth and development, organ dysfunction, and secondary malignancies [12,13]. Therefore, improving complete resection rates is of paramount importance.

However, margin assessment is difficult due to the infiltrative and diffuse growth pattern of ES. Furthermore, preoperative imaging modalities, such as MRI and CT, which are used to identify the tumor location and extension, cannot be directly translated to the intraoperative setting for various reasons. First, these images are limited to a two-dimensional space, whereas the surgery is performed in a three-dimensional environment. Secondly, they do not precisely overlap with the intraoperative situation due to tissue manipulation and positioning. Hence, surgeons have to rely mostly on their tactile and visual feedback, which hinders intraoperative identification of the tumor border. As a consequence, incomplete resections occur in 20–30% of the ES patients [8,14].

Fluorescence-guided surgery (FGS) is an emerging tool that could help to increase the chance of complete tumor resections [15]. It has been developed to assist surgeons in accurately identifying critical anatomical structures, including tumors. By administering a fluorescent tracer, FGS enables real-time visualization of the structure of interest, facilitated by a dedicated camera system. FGS can be categorized into non-targeted and targeted (tumor-specific) FGS, which has been recently reviewed for pediatric ES [16]. Targeted FGS involves the use of tumor-specific tracers, such as antibodies or peptides conjugated to a fluorophore, to selectively recognize and bind to targets overexpressed on tumor cells. Currently, targeted FGS is being explored for various tumor types and has shown promising outcomes, presenting a potential avenue for improving ES surgery [17,18,19,20,21].

Targeted FGS can be implemented in two distinct ways: through intravenous (IV) administration of a near-infrared (NIR) fluorescent tracer or via topical application of a fluorescent tracer onto the resected specimen. Each method possesses its own advantages and disadvantages. The IV administration of a NIR fluorescent tracer is considered the preferred modality due to its capability of providing real-time intraoperative fluorescence assessment of both malignant and surrounding healthy tissue, including the detection of possible satellite lesions in the tumor bed. However, the development of a NIR fluorescent tracer entails considerable costs and time investment. It necessitates comprehensive evaluation of toxicology and safety, determination of optimal dosage, window of imaging, and subsequent registration as an approved drug. Moreover, certain tracers may never become available due to significant side effects resulting from binding to healthy (distant) tissues. FGS by ex vivo topical application, on the other hand, can be a relatively feasible step towards the development of a FGS tracer for IV use, as it circumvents the need for expensive and time-consuming toxicology studies for human application [16]. Although the detection of potential satellite lesions is not possible, topical application can still yield clinical benefits by aiding in the intraoperative assessment of tumor margins. Furthermore, this method enables the analysis of fluorescence within the patient-specific tumor-associated stromal and vascular composition and facilitates the evaluation of inter- and intratumoral heterogeneity. Thus, it may reduce the reliance on animal experiments, which should be minimized by EC regulation, but also because they may not accurately reflect the human background fluorescence necessary for calculating the tumor-to-background ratio (TBR).

Three important parameters define successful targeted FGS: the tumor-specific biomarker (target), the targeting moiety conjugated to a fluorophore (tracer), and the dedicated camera system (of which several are already on the market) [22]. Encouragingly, Bosma et al. already identified some promising targets for ES, based on their overexpression on ES tumor tissue as assessed by immunohistochemistry (IHC) or flow cytometry [23]. The following targets were identified: Cluster of Differentiation 99 (CD99), C-X-C chemokine receptor type 4 (CXCR4 also known as CD184), Cluster of Differentiation 117 (CD117, also known as C-Kit), Neuropeptide Y receptor type 1 (NPY-R-Y1), Leucine rich repeat, Immunoglobin-like domain-containing protein 1 (LINGO-1), and insulin-like growth factor 1 receptor (IGF-1R) [23]. Additionally, disialoganglioside 2 (GD2) could be a promising target due to its known expression of 40–90% in ES biopsy samples and the clinical availability of an FGS tracer for neuroblastoma, which will soon be tested in a phase 1/2 clinical trial [24,25,26,27].

However, while these potential targets have shown overexpression in pediatric ES, their expression on healthy tissue surrounding the tumor has not been assessed. As a crucial step towards developing targeted FGS for ES, this study aims to evaluate the selected targets through IHC analysis conducted on biopsies and resection specimens, including adjacent healthy tissue. Including resection specimens with adjacent healthy tissue allows the evaluation of target expression after preoperative treatment and assessment of the differential expression between healthy and malignant tissue which will affect the TBR. Subsequently, three tracers with the highest potential for FGS were further evaluated with fluorescent tracers. Binding specificity of the tracers was assessed using cell-line-based flow cytometry and immunofluorescence microscopy experiments. Finally, as proof-of-concept, a topical application experiment with the three selected tracers was performed on freshly resected tissue from pediatric ES tumor and surrounding healthy tissue.

## 2. Materials and Methods

### 2.1. Sample Selection

ES patients who gave informed consent for the use of their tissue at the time of surgery and underwent surgical resection between 2018 and 2022 were eligible for this study. Biopsies and sections of ES tissue samples with tumor and adjacent healthy tissue were selected by a sarcoma pathologist. Subsequently, all available corresponding formalin-fixed paraffin-embedded (FFPE) biopsy material and resection specimens were collected from the biobank of the Princess Maxima Center for Pediatric Oncology, The Netherlands. Ethical approval for collecting FFPE tissue and for the use of freshly excised tissue was acquired from the institutional review board of the Princess Maxima Center for Pediatric Oncology (PMCLAB2022.317). The study was conducted in accordance with the Declaration of Helsinki.

### 2.2. Biomarkers

Seven biomarkers were selected for IHC evaluation based on their potential for FGS in ES, as reported in the literature [23]. The biomarkers CD99, CXCR4, CD117, NPY-R-Y1, LINGO-1, and IGF-1R were based on a systematic review that selected promising targets for FGS in ES [23]. GD2 was added for its known overexpression on ES and the availability of a clinically targeted tracer [23,24,25].

### 2.3. Immunohistochemistry 

FFPE biopsies from 13 ES patients, and 8 corresponding resection specimens with adjacent healthy tissue, were included in the IHC evaluation. All of these selected cases received preoperative chemotherapy as standard treatment at the Princess Maxima Center for Pediatric Oncology (Table 1). Sections from FFPE blocks having a 4 µm thickness were cut, mounted on pre-coated slides, and dried for at least 30 min at 70 °C. For IGF-1R, sections were manually deparaffinized in xylene for 15 min, rehydrated in a series of 100%, 50%, and 25% ethanol dilutions, and rinsed in demineralized water. Endogenous peroxidase was blocked with 0.3% hydrogen peroxide in demineralized water for 20 min. Antigen retrieval was performed using the PT Link (Dako, Glostrup, Denmark; Appendix A), and two washing steps of 5 min in phosphate buffered saline (PBS) were performed. Sections were incubated in a humidified room with 150 μL primary antibody using predetermined solutions based on test stainings on control tissue (Appendix A). Afterwards, slides were washed three times in PBS for 5 min and incubated with the appropriate secondary goat-anti rabbit HRP conjugate (catalog number K4003, Agilent Technologies, Santa Clara, CA, USA), followed by an additional washing step. Staining was visualized through incubation with 3,3-diaminobenzidine tetrahydrochloride solution (DAB, K3468, Agilent Technologies, Inc., Santa Clara, CA, USA) for 10 min at room temperature. For the other biomarkers, deparaffinization and IHC stains were performed with an automated BOND-RX system (Leica Microsystems, Wetzlar, Germany). Antigen retrieval was performed by boiling the sections in TRIS/EDTA (BOND Epitope Retrieval Solution 2, pH9; Leica Biosystems, Wetzlar, Germany) for 15 min for CXCR4, GD2, and LINGO-1, for 20 min for CD99 and CD117, and for 30 min for NPY-R-Y1. Primary and secondary antibodies used for IHC evaluation were selected based on the literature (Appendix A). Endogenous peroxidase was blocked with 0.3% hydrogen peroxide in demineralized water for 5 min and the sections were incubated for 15 min at RT with the primary antibodies, except for LINGO-1 which was incubated for 30 min at RT. The sections were then incubated for 8 min with a post primary rabbit anti-mouse linker followed by incubation for 8 min with anti-rabbit horseradish-peroxidase-labeled polymer. After incubation for 10 min with diaminobenzidine, all slides were counterstained for 5 min with hematoxylin (BOND Polymer Refine Detection Kit; Leica Biosystems), dehydrated, cleared, and mounted. On each slide a positive and negative control tissue was included (identified with the human protein atlas; Appendix A) [28]. Stained sections were digitalized with the Aperio Scanner (Leica Biosystems, Wetzlar, Germany) and viewed with ImageScope Software (version 12.4.6, Aperio ePathology, Leica).

### 2.4. Immunohistochemistry Scoring Method

IHC scoring was performed by a pathologist specialized in sarcomas. For assessment, an ordinal scale was used based on the percentage of stained cells and staining intensity (number of stained cells: 0 ≤ 10%; 1 = 10–25%; 2 = 25–50%; 3 = 50–75%; and 4 ≥ 75%; staining intensity: 0 = no staining; 1 = mild; 2 = moderate; 3 = strong). The two scores were then multiplied to create the final immunoreactive score (IRS) with a range from 0 to 12. The final expression score was the IRS subdivided into 3 categories (0–3 = weak expression; 4–8 = moderate expression, and 9–12 = strong expression). An example of the immunohistochemical scoring method in biopsies of ES tumors is depicted in Figure 1. Biomarkers with strong expression in the tumor and weak expression in adjacent healthy tissue were considered most suitable for FGS. 

### 2.5. Human Cancer Lines

ES cell lines A673 and RD-ES were selected as cells to assess binding of the tracers, while neuroblastoma cell lines SK-N-BE and KCNR, and breast cancer organoid lines BC27T and BC62T served as positive or negative controls for the selected targets (Table 2). Controls were selected based on RNA expression for the targets of interest (Appendix B). ES cell lines were obtained from American Type Culture Collection (ATCC, Manassas, VA, USA), neuroblastoma cell lines from cryopreservation at the Princess Maxima Center, The Netherlands, and breast cancer organoid lines were developed, with informed consent from all donors at Hubrecht Organoid Technology, Utrecht, The Netherlands. Short tandem repeat (STR) profiling was performed to ensure quality and integrity of the ES cell lines. A673 cells were cultured in Dulbecco’s modified Eagle’s medium (DMEM) containing high glucose, glutamine, phenol red, and sodium pyruvate (Gibco, Invitrogen, Carlsbad, CA, USA), 10% Fetal Bovine Serum (FBS; Hyclone, Thermo Scientific, Rockford, IL, USA), and penicillin/streptomycin (PS, both 100 IU/mL; Invitrogen). RD-ES cells were cultured in RPMI-1640 medium (Invitrogen) with 10% FBS and PS. SK-N-BE and KCNR cells were cultured in DMEM4x medium, which consists of DMEM (Invitrogen) supplemented with high glucose, 2 mM L-glutamine (Invitrogen), Minimum Essential Medium (MEM) non-essential amino acids (Invitrogen), 10% FBS and PS. BC27T and 62T cells were received in 12-well suspension plates (Greiner Bio-One GmbH, Firckenhausen, Germany) seeded in basement membrane extract (Cultrex BME, R&D Systems, Minneapolis, MN, USA) just before the experiment was conducted [29,30]. Absence of mycoplasma in all cell lines was confirmed using polymerase chain reaction. Cells were grown to 90% confluence in a humidified incubator at 37 °C (5% CO_2_) and detached with trypLE/EDTA (Invitrogen). 

### 2.6. Antibodies Used for Flow Cytometry and Immunofluorescence Microscopy

Mouse IgG2a kappa monoclonal CD99 antibody (3B2/TA8) labeled with FITC and mouse IgG1 kappa monoclonal CD117 antibody (YB5.B8) labeled with PE were bought from Invitrogen. Mouse IgG2a kappa isotype control (eBM2a) labeled with FITC and mouse IgG1 kappa isotype control (P3.6.2.8.1) labeled with PE (Invitrogen) were recommended (by Invitrogen) and used as isotype controls. A chimeric monoclonal antibody against GD2 (Dinutuximab-beta, Qarziba, Laupheim, Germany) was conjugated to Alexa Fluor™ NHS ester 647 (Invitrogen). The Alexa Fluor (AF) NHS esters were dissolved in anhydrous dimethylsulfoxide (DMSO) (Invitrogen, D12345, Carlsbad, CA, USA), and the reaction was carried out in 0.5 M Hepes buffer (15630–056, Gibco, Carlsbad, CA, USA), pH 8.0, at room temperature for two hours. 0.1 M Tris was added to quench the reaction. The antibody-fluorophore conjugate was purified twice using a gel filtration column (Zeba Spin Desalting Column, 40 MWKO, Thermo Fisher Scientific, Waltham, MA, USA). The degree of labeling (DoL) was calculated by measuring the protein concentration and fluorophore concentration using the NanoDrop™ One (Thermo Fisher Scientific, Waltham, MA, USA). A DoL around 1–1.5 was considered successful as this is generally recommended for FGS probes [31].

### 2.7. Flow Cytometry

After detachment, cells were adjusted to 0.3 × 106 viable cells/well in FACS buffer (10% FBS, ThermoFisher, Waltham, MA, USA) in a 96-well U-bottom plate (Invitrogen). Cells were incubated on ice for 45–60 min, avoiding light exposure, with 5 μg/mL 3B2/TA8-FITC, 5 μg/mL YB5.B8-PE, and 1 μg/mL Dinituximab-AF647 in FACS buffer containing live/dead staining (LIVE/DEAD™ Fixable Near-IR Dead Cell Stain, ThermoFisher). Apart from negative control cell lines, isotype controls (5 μg/mL eBM2a-FITC and 5 μg/mL P3.6.2.8.2-PE) were added to the cells in separate wells to assess binding specificity. Next, cells were washed twice, and flow cytometry measurements were performed using the CytoFLEX LX (Beckman Coulter, Brea, CA, USA) using the lasers and detectors to measure FITC, PE, AF647, and the live/dead staining (respectively 525/40, 585/42, 695/40, and 763/43). Analysis was performed using FlowJo software (TreeStar, Woodburn, OR, USA). 

### 2.8. Immunofluorescence Microscopy

After detachment, cells were adjusted to 0.3 × 106 viable cells/well in culture medium in a Greiner 96-well glass-bottom plate (Sigma-Aldrich, St. Louis, MO, USA). Cells were incubated for 45–60 min in a humidified incubator at 37 °C and 5% CO_2_ with similar antibody concentrations as described for flow cytometry. In addition, a cell membrane staining CellBrite 450 (Biotium, Fremont, CA, USA) was added 1:1000 simultaneously with the antibodies. Next, imaging was performed with the SP8 Leica microscope (Leica Microsystems, Wetzlar, Germany) using a HC PL APO 10X/0.40 CS2 objective with zoom set to 1.0 and digitized in 16 bits per voxel. The 488 nm, 561 nm, and 633 nm lasers were used for FITC, PE, and AF647, respectively. For detection of every tracer, a separate photo-multiplier tube (PMT) detector was used with the minimum and maximum wavelengths set according to the reference spectra of FITC, PE, and AF647. Three-dimensional rendering was performed using Imaris x64 (10.0 Bitplane).

### 2.9. Topical Application and Imaging

Freshly resected tumor tissue and adjacent healthy (bone and muscle) tissue were derived from the proximal femur of a pediatric ES patient directly after surgery. A pathologist specialized in sarcoma, blinded for fluorescence, selected tissue samples from clinically relevant regions (with tumor and adjacent healthy tissues) based on visual inspection of the slides of the resection specimen. The acquired tissue was then cut into slices of approximately 0.5 cm × 0.5 cm × 0.2 cm. These tissues were immediately incubated for 45–60 min in a humidified incubator at 37 °C and 5% CO_2_ with CellBrite 450, 3B2/TA8-FITC, YB5.B8-PE, and Dinituximab-AF647 using the same concentrations as described for flow cytometry. Directly after incubation without any washing steps, multispectral imaging of the specimen was performed with the SP8 Leica microscope (Leica Microsystems, Wetzlar, Germany) using a HC PL APO 10X/0.40 CS2 objective with zoom set to 1.0 and digitized in 16 bits per voxel. Three-dimensional rendering was performed using Imaris x64 (10.0 Bitplane).

### 2.10. Statistical Analysis

Each biomarker’s categorical IRS (0–3 = weak expression; 4–8 = moderate expression, and 9–12 = strong expression) was depicted with bar charts created by GraphPad Prism 9 (La Jolla, CA, USA).

## 3. Results

### 3.1. Immunohistochemistry

In total, 13 patients (8 biopsies with corresponding post-chemotherapy resection specimen and 5 biopsies without) were included for the IHC evaluation. This cohort had a median age of 11 years (range: 2–19 years), most patients were male (n = 8/13; 62%), and all patients received preoperative chemotherapy (Table 1). IHC analysis showed high percentages of stained ES cells for CD99, CD117, and NPY-R-Y1, mostly with moderate/strong intensity, while expression on adjacent healthy tissue was limited (Figure 2). The tumor boundary was determined by a pathologist specialized in sarcoma using hematoxylin eosin (H&E) stained samples. The immunoreactive score was not utilized for differentiation between tumor and healthy tissues. Occasionally, false positive signals were observed in healthy tissue, posing a risk of over-resection and potential loss of function. However, CD99 and CD117 showed relatively lower expression in healthy tissue compared to the tumor tissue. Despite some inter-tumor variability, both CD99 and CD117 were consistently expressed in both tumor biopsy and tumor resection specimens. For CD117, the average IRS score in post-chemotherapy resection specimen was lower compared to biopsies, which could mean that CD117 expression decreased after chemotherapy. For the other biomarkers, we did not observe this phenomenon as IRSs were comparable in the biopsy and corresponding resection specimen (Figure 3). For GD2 and LINGO-1, we could not establish a working IHC protocol for the acquired FFPE tissue.

### 3.2. Flow Cytometry and Immunofluorescence Microscopy

CD99 and CD117 were selected for cell line-based experiments based on their high IRS scores in ES tumor, but limited expression in adjacent healthy tissue. In addition, GD2 was investigated since an FGS tracer for this biomarker is in the developmental phase and previous literature reported relatively high GD2 expression (40–90%) in ES tissue [24,25,26]. First, binding of anti-CD99 (3B2/TA8-FITC), anti-CD117 (Yb5.B8-PE), and anti-GD2 (Dinutuximab-AF647) was evaluated with flow cytometry. A673, RD-ES, BC62T, and SK-N-BE cells showed respectively high, high, moderate, and almost negative binding of anti-CD99 (Appendix C). A673, RD-ES, BC62T, and BC27T cells showed respectively low, moderate, moderate, and negative binding of anti-CD117 (Appendix C). A673, RD-ES, KCNR, and BC27T cells showed respectively low, moderate, high, and low binding of anti-GD2 (Appendix C). Next, immunofluorescence microscopy confirmed flow cytometry results; membranous binding of anti-CD99 and anti-CD117 was observed for all ES cells, while anti-GD2 only stained some ES cells (Figure 4).

### 3.3. Topical Application

Fresh tissue from a proximal femur resection specimen was collected directly after surgery (Figure 5). Tumor and adjacent healthy (bone and muscle) tissues were immediately incubated with CellBrite 450, anti-CD99, anti-CD117, and anti-GD2. Topical application of anti-CD99 and anti-GD2 on ES tumor showed fluorescence, while anti-CD117 did not. None of the tracers was fluorescent on neighboring healthy muscle or bone tissues (Figure 6).

## 4. Discussion

In this study, we first evaluated ES specific biomarkers on FFPE tumor and adjacent healthy tissue with IHC. Based on their IRS, CD99 and CD117 were considered the most promising targets for FGS in ES. In addition, GD2 was added as a promising biomarker based on previously reported high expression in ES and the exceptional availability of a clinical FGS tracer for pediatric patients (currently in clinical phase 1/2 trial for neuroblastoma) [25]. Next, flow cytometry and fluorescence microscopy experiments with fluorescent antibodies targeting CD99, CD117, and GD2 revealed, respectively, high, low/moderate, and low expression on living ES cells. Then, as proof-of-concept, an ex vivo topical application experiment was conducted on one freshly resected ES tumor and adjacent healthy tissue. Although not generalizable due to the small sample size, CD99 and GD2 targeting tracers showed fluorescent signal on the ES tumor, whereas anti-CD117 did not show fluorescence for this patient. Importantly for the intended application, none of the tracers showed fluorescence on adjacent healthy (bone and muscle) tissue. 

We adhered to standard preclinical workup for the evaluation of biomarkers for targeted FGS. After the selection of potential targets by Bosma et al., we conducted IHC experiments including both healthy and malignant tissue [32]. While LINGO-1 and GD2 have demonstrated high expression in ES (40–91%), their expression in adjacent healthy tissue has not been characterized [24,33]. Unfortunately, we encountered difficulties in establishing a working IHC staining protocol for LINGO-1 and GD2 on FFPE tissue. GD2 is of particular interest due to the availability of a clinical tracer [25]. High GD2 expression has been reported in various pediatric tumor types, but inconsistencies in measured levels via IHC have been observed [24,25,34]. This discrepancy can be attributed to the solubility of GD2 in certain solvents, such as ethanol, which is required for FFPE tissue preparation [27,34]. Fresh frozen tissue could serve as an alternative, but flow cytometry on living cells also overcomes the limitations caused by sample processing [27,35]. Notably, Wingerter et al. employed flow cytometry to evaluate GD2 expression and, in line with our findings, reported low GD2 expression levels in the same commercially available ES cell lines (A673 and RD-ES) [27]. However, the percentage of GD2-positive cells in six primary (non-commercially available) ES cell lines ranged from low (1%) to high (98%) [27]. The heterogeneity in GD2 expression in ES cells could be attributed to the degree of tumor cell differentiation [28]. Therefore, GD2 expression might be specific to individual patients rather than tumor entities. Despite the low binding of anti-GD2 to ES cell lines A673 and RD-ES, as determined by flow cytometry and fluorescence microscopy, anti-GD2 exhibited a fluorescent signal in our topical application experiment. Together with the known heterogeneous GD2 expression in ES, this suggests that it should not be disregarded as a potential candidate for FGS.

We consider CD99 the most promising target for FGS in ES due to its consistent high tumor-specific expression demonstrated in both in vitro and ex vivo experiments. As emphasized in the introduction, the ex vivo topical application of a CD99-targeting tracer on resected tumor specimens can serve as a practical initial approach for intraoperatively assessing tumor margins. This approach offers advantages such as bypassing the time-consuming and costly development of an intravenously administered tracer, while at the same time providing clinical benefits by aiding in intraoperative tumor margin assessment. Successful translation to IV use can be anticipated once ex vivo topical application consistently yields promising results, as demonstrated with EMI-137 targeting c-MET in penile squamous cell carcinoma [36]. However, multiple essential aspects need to be assessed before clinical translation of a targeted tracer, such as comprehensive evaluation of biodistribution, toxicology, determination of optimal dosage and safety, and subsequent registration as an approved drug. It is worth noting that fully humanized monoclonal antibodies and single-chain variable fragment (scFv) against CD99 are already available and have exhibited minimal toxicity to healthy peripheral blood cells [37]. Ideally, the smaller anti-CD99 scFv should be transformed into a clinical tracer by conjugation to a NIR fluorophore. This approach would enable the achievement of a high TBR within hours after IV administration, compared to days required for large-sized antibodies [16,37]. 

Ex vivo topical application experiments offer an additional advantage by enabling the simultaneous testing of multiple tracers when they are conjugated with fluorophores that have significantly different excitation and emission spectra. Considering tracers targeting CD117 and GD2 as potential alternatives to CD99, these could be included in the same trial. However, it is important to consider that the emission wavelength of the fluorophore may affect the penetration depth of the fluorescent signal and the level of autofluorescence. If the objective is to test only one tracer per patient using topical application, or when multiple tracers become available for IV use in the future, biopsies can be utilized to select the most promising patient-specific tracer. Our findings indicate that, except for CD117, the expression of the tested tracers for ES was relatively similar between biopsy and resection specimens. In cases where single tracers are unable to fully cover the tumor with fluorescence due to intratumoral heterogeneity, a possible solution would be to evaluate a combination of tracers [32]. 

This study has some limitations that should be acknowledged. Firstly, the rarity of ES resulted in a limited availability of FFPE samples. Additionally, despite our efforts, we were unable to establish a working ICH protocol for LINGO-1 and GD2 on the available FFPE tissue. In addition, comparing IHC stainings for the purpose of FGS should be undertaken with caution due to the variability of results depending on decalcification methods and the type of antibodies, dilutions, epitopes, clones, or staining protocols used [38]. Moreover, the IRS scoring can be seen as subjective due to intra- and inter-observer variability, and objective scoring methods using semi-automated imaging software such as ImageJ or QuPath could improve the quality of this study [39,40,41]. However, evaluation with QuPath required a test and validation set, which was unfeasible for our small cohort of ES samples. Additionally, scoring of our pathologist specialized in sarcoma (working in a tertiary referral center) is currently seen as the gold standard. Therefore, no further objective scoring was performed. Although we focused on the most promising targets, namely CD99, CD117, and GD2, we also found NPY-R-Y1 to be of interest and worthy of further investigation. Furthermore, the topical application was performed on only one freshly resected ES bone specimen. The applicability for FGS of ES-based tracers has to be confirmed in a larger cohort of pediatric ES patients, preferably in a multicenter setting. This study was performed to showcase the proof-of-principle, and to encourage future studies to conduct topical application experiments with a large cohort of ES patients. Yet, the feasibility of targeted FGS in a large number of bone and soft tissue ES patients, as well as the possible applicability in non-Ewing SRCT, remains to be investigated. Ideally, FGS would be able to identify both bone and soft tissue margins. However, mainly the soft tissue expansion and the periosteal or subperiosteal expansion would be useful for visualization with FGS, as the osseous tumor margin could also be navigated with X-ray or computer-assisted surgery [42]. Nonetheless, with promising results obtained from IHC, in vitro experiments using two patient-derived ES cell lines (A673 and RD-ES), and a proof-of-principle experiment conducted on freshly resected ES tissue, we believe that this study demonstrates the potential of FGS for improving ES treatment in the future.

## 5. Conclusions

Based on the immunohistochemical evaluation, preclinical in vitro experiments, and ex vivo topical application experiment on freshly resected Ewing sarcoma and adjacent healthy tissue, our findings suggest that CD99-targeting tracers hold promise for fluorescence-guided surgery of Ewing sarcoma. Furthermore, tracers targeting CD117 and GD2 show potential as alternative options. As a next step towards the clinical implementation of intravenous tracers for targeted fluorescence-guided surgery in Ewing sarcoma, we recommend conducting an ex vivo topical application study on a large cohort of Ewing sarcoma patients. This study would provide valuable insights and pave the way for further advancements in the field of Ewing sarcoma treatment. 

## Figures and Tables

**Figure 1 cancers-15-03896-f001:**
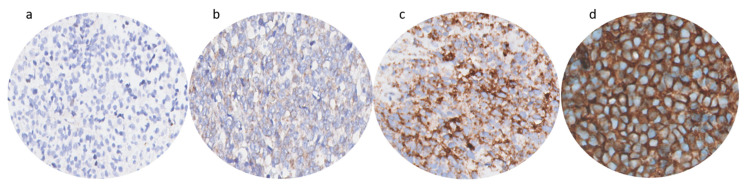
Example of immunohistochemical scoring method in biopsies of ES tumors. An ordinal scale was used based on the percentage of stained cells and staining intensity. The two scores were then multiplied to create the final immunoreactive score (IRS); (**a**) ≤10% stained cells (=score 0) and no expression intensity (=score 0) creates an IRS of 0 (IGF-1R); (**b**) =10–25% stained cells (=score 1) and mild intensity (=score 1) creates an IRS of 1 (CXCR4); (**c**) ≥75% stained cells (=score 4) and moderate intensity (=score 2) creates an IRS of 8 (CD117); (**d**) ≥75% stained cells (=score 4) and strong intensity (=score 3) creates an IRS of 12 (CD99).

**Figure 2 cancers-15-03896-f002:**
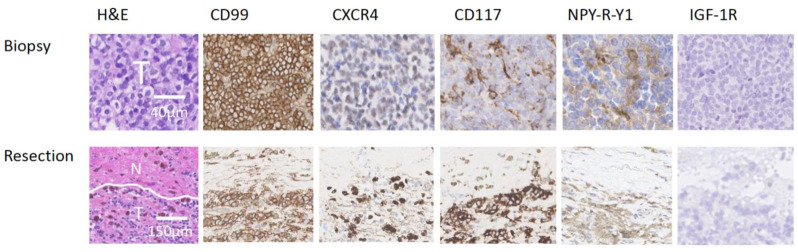
Representative IHC staining of biopsy and corresponding post-chemotherapy resection specimen from an ES patient. The H&E staining shows tumor cells in biopsy material (T), and adjacent healthy “normal” tissue (N) as well as tumor tissue (T) in resection material. Overall, CD99, CD117, and NPY-R-Y1 showed high percentages of stained ES cells in biopsy and resection material, with moderate to strong intensity, while staining in adjacent healthy tissue was limited. In this case CXCR4 displayed a moderate percentage of stained cells with mild to strong intensity, while IGF-1R did not show any stained cells.

**Figure 3 cancers-15-03896-f003:**
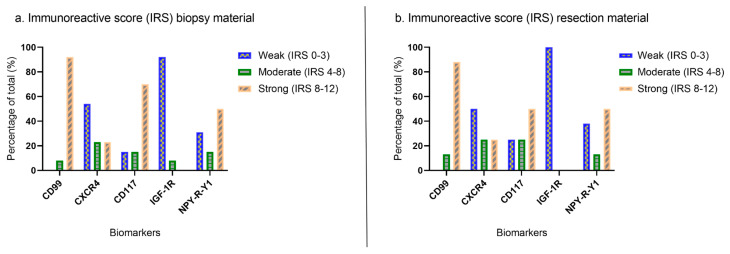
Immunoreactive score (IRS) of the selected biomarkers divided into three categories; weak (IRS 0–3), moderate (IRS 4–8), and strong (IRS 8–12). Separate results are depicted for (**a**) biopsy material (n = 13) and (**b**) resection specimen (n = 8).

**Figure 4 cancers-15-03896-f004:**
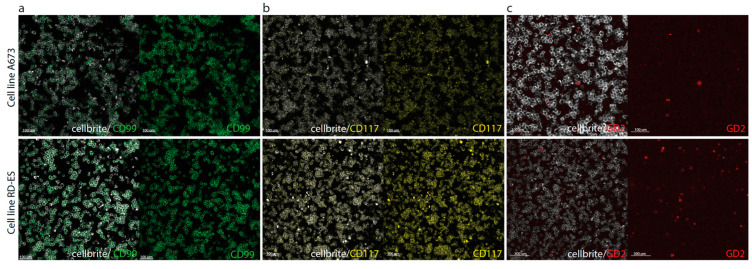
Immunofluorescence analysis of (**a**) 3B2/TA8-FITC (anti-CD99); (**b**) Yb5.B8-PE (anti-CD117); and (**c**) Dinutuximab-AF647 (anti-GD2) binding to A673 and RD-ES cells. Anti-CD99, anti-CD117 and anti-GD2 are shown in green, yellow, and red, respectively. The cell membrane marker CellBrite 450 stained all cell membranes (white).

**Figure 5 cancers-15-03896-f005:**
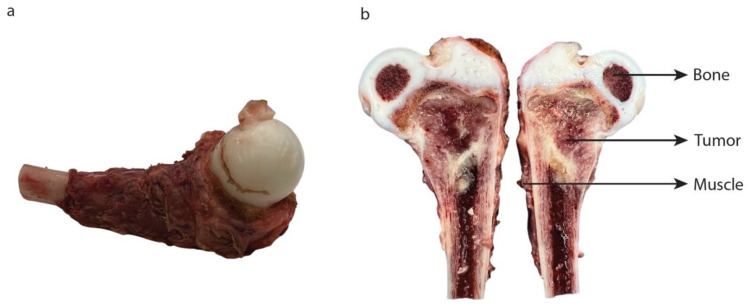
Freshly resected ES: (**a**) left proximal femur of a 2-year-old child; (**b**) bisected specimen shows fresh tumor tissue with adjacent healthy bone and muscle (used for the topical application experiment).

**Figure 6 cancers-15-03896-f006:**
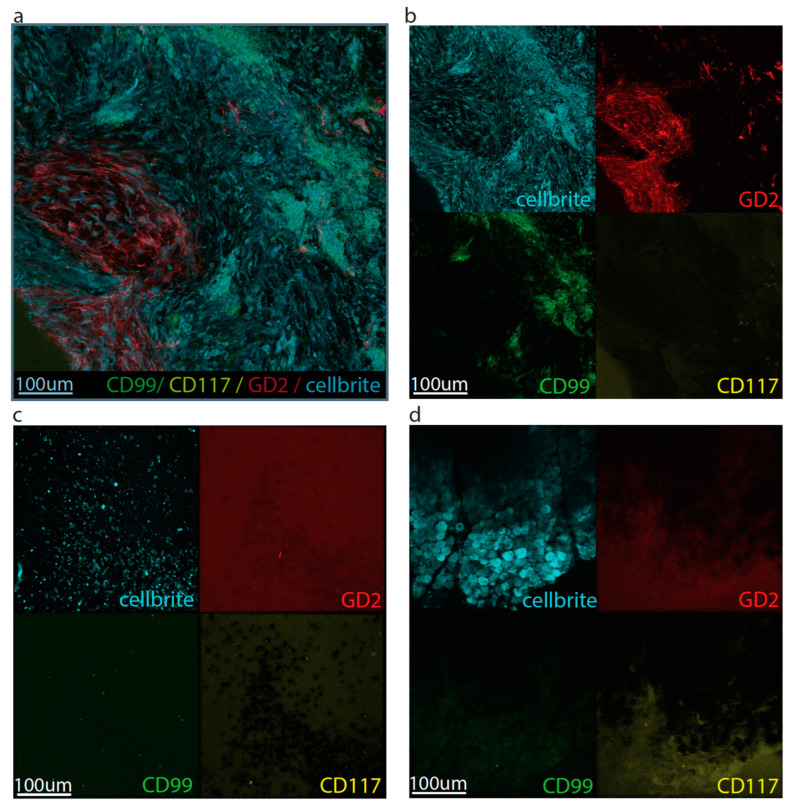
Immunofluorescence evaluation of 3B2/TA8-FITC (anti-CD99, green), Yb5.B8-PE (anti-CD117, yellow), and Dinutuximab-AF647 (anti-GD2, red) binding to a freshly resected ES tumor (**a**,**b**) and adjacent healthy bone (**c**) and muscle (**d**) tissues ex vivo. Anti-CD99 and anti-GD2 showed fluorescent signal in the tumor explant, while anti-CD117 did not show fluorescent signal. No fluorescence was observed in adjacent healthy bone and muscle tissue. The cell membrane marker (CellBrite 450) clearly stained all cell membranes (cyan).

**Table 1 cancers-15-03896-t001:** Ewing sarcoma patient and tumor characteristics.

Patient	Gender	Age *	Preoperative Therapy		% Vital Tumor in Resection Specimen	cTNM
1	Female	13	VIDE		30%	cT1N0M0
2	Female	16	VIDE, Melphalan/Treosulfan, Irinotecan/Temozolomide		N.A.	cTxN0M0
3	Male	14	VIDE, Dactinomycin, Melphalan/Treosulfan		40%	cT2N1M1
4	Female	2	VIDE, VAI		N.A.	cT1N0M1
5	Male	5	VIDE, VAI		40%	cT1N0M0
6	Male	14	VIDE, VAC, Melphalan/Treosulfan		40%	cT2N1M0
7	Male	6	VIDE, VAI, Busulfan/Melphalan		80%	cT1N0M0
8	Male	5	Cyclofosfamide/Topotecan, Tremozolamide/Irinotecan, Treosulfan/Melphalan, VIDE, VAI, Vinorelbine		N.A.	cT2N0M1
9	Male	6	VIDE, VAI		N.A.	cT2N0M0
10	Female	8	VIDE, Treosulfan/Melphalan		40%	cT2N1M0
11	Female	18	VIDE		N.A.	cT2N0M0
12	Male	16	VIDE, VAI, Treosulfan/Melphalan		50%	cT2N0M0
13	Male	14	VIDE, Treosulfan/Melphalan, VAC, Cyclofosfamide/topotecan		40%	cT2N0M1

* Age at diagnosis in years. Abbreviations: VIDE = Vincristine Ifosfamide Doxorubicin Etoposide; VAI = Vincristine Dactinomycin Ifosfamide; VAC = Vincristine, Doxorubicin, Cyclophosphamide; N.A. = Not applicable, since only biopsy material but no resection specimen was available.

**Table 2 cancers-15-03896-t002:** Human cancer lines selected as tested and positive and negative control lines for the three selected targets, CD99, CD117, and GD2.

Target	Tested Ewing Sarcoma Cell Lines	Positive Control	Negative Control
CD99	A673 and RD-ES	BC62T	SK-N-BE
CD117	A673 and RD-ES	BC62T	BC27T
GD2	A673 and RD-ES	KCNR	BC27T

## Data Availability

The data presented in this study are available on request from the corresponding author. Some of the data are not publicly available due to confidentially and in accordance with the Dutch Personal Data Protection Act.

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
