# Peer review of "Evaluation of Potential Targets for Fluorescence-Guided Surgery in Pediatric Ewing Sarcoma: A Preclinical Proof-of-Concept Study"

_cancers, 2023, doi:10.3390/cancers15153896_

Round 1

Reviewer 1 Report

Abstract

1. While the abstract suggests that anti-CD117 does not show fluorescence for the patient in question, it also indicates that CD117 could be a potential alternative tracer

2. Line 42 - the use of the term "clinically available tracer" might be misleading. GD2 is a biomarker, and the abstract later refers to "anti-GD2", which would be a specific tracer. It's important to differentiate between the marker and the tracer designed to bind to it.

3. The abstract lacks a clear statement of the research question or objective. It would be useful to provide a sentence explaining the purpose of the study to provide context for the findings.

Introduction

The introduction might benefit from a brief discussion of the relevance and significance of FGS in pediatric ES specifically. The text explains the general utility of FGS, but a brief explanation of why it is particularly important or useful in pediatric ES could help to ground the paper's study in a specific context.

1.  The sentence "Fluorescence-guided surgery (FGS) is an emerging tool that could assist surgeons to achieve complete tumor resections" could benefit from specifying that FGS can help increase the chances of complete tumor resections. As it currently stands, it may give the impression that FGS guarantees complete resection, which is not necessarily true.

2. "targeted FGS is being explored for various tumor types and has shown promising outcomes" - cite some specific studies that demonstrate these promising outcomes

3. Line 97-98 - "FGS by ex vivo topical application on the other hand, can be a relatively easy step towards the development of a FGS tracer for IV use..." - the term 'easy' might not correctly represent the challenges and complexities involved in developing a FGS tracer for IV use. I suggest replacing this term with 'feasible' in order for it to be a more appropriate term here.

Methods

They are well written and well described.

Results

The Figures and Images are well presented and clear.

Discussion

1. In the phrase "an ex vivo topical application experiment was conducted on one freshly resected ES-tumor and adjacent healthy tissue. CD99 and GD2 targeting tracers showed fluorescent signal on the ES-tumor, whereas anti-CD117 did not show fluorescence for this patient." the sample size is extremely small. One cannot generalize findings from a single experiment on one patient. This limitation should be more strongly emphasized.

References

Ok

There are some minor phrasing errors. I've mentioned some of them in the above comments.

Author Response

Dear Reviewer 1,

We sincerely appreciate all the reviewer’s valuable comments and suggestions, which helped us in improving the quality of the manuscript.

On behalf of the coauthors,

Zeger Rijs

Reviewer 2 Report

this study aims to evaluate the selected targets through IHC analysis conducted on biopsies and resection specimens, including adjacent healthy tissue. 

The study is well designed and the paper well organized.

Why was the IHC analysis performed  on post cht tissues in all cases? On the surgical specimen, how were samples on which doing ICH analysis selected?

Were all ES bone sarcomas?Would this method feasible also in ES of soft tissues? Also, would it be applicable to non Ewing SRCT?

Please discuss further about possible practical application. Would it be more useful to identify bone or soft tissue margins? 

Author Response

Dear Reviewer 2,

We sincerely appreciate all the reviewer’s valuable comments and suggestions, which helped us in improving the quality of the manuscript.

On behalf of the coauthors,

Zeger Rijs

Reviewer 3 Report

Fluorescence-guided surgery (FGS) is one of the promising technological advances facilitating the visualization of tumors in real-time during surgery. Based on Bosma et al. identification, the authors selected CD99,CD117,CXCR4, GD2 and IGF-1R as the potential targets, and concluded that CD99-targeting tracer could be beneficial for pediatric ES patients.  Specific comments are listed below.

1. We noticed that  in Fig 2. there are some positive expression of CD99, CD117 in the authors-claimed healthy tissue in resected speciments. How does the author draw the boundary between tumor and healthy "normal"tissue? Is there a cut-off ratio of immunoreactive score to differentiate the tumor and the "normal" tissue? This is important because false positive signals in the healthy tissue may cause unnecessary resection of tissue and loss of function in the future clinical use

2. Although tracers can be topically applied to freshly resected tumor specimen, the disadvantage of this method is unable to detect the possible satellite lesion in the tumor bed, compared to IV approach.

3. Small sample size is inevitably weaken the level of evidence. The applicability for fluorescence-guided surgery of ES based tracers has to be confirmed in a larger cohort of pediatric ES patients in a multicenter setting.

Author Response

Dear Reviewer 3,

We sincerely appreciate all the reviewer’s valuable comments and suggestions, which helped us in improving the quality of the manuscript.

On behalf of the coauthors,

Zeger Rijs
